# Glial granules contain germline proteins in the *Drosophila* brain, which regulate brain transcriptome

Samuel J. Tindell[1], Eric C. Rouchka [2] & Alexey L. Arkov [1✉]

Membraneless RNA-protein granules play important roles in many different cell types and organisms. In particular, granules found in germ cells have been used as a paradigm to study large and dynamic granules. These germ granules contain RNA and proteins required for germline development. Here, we unexpectedly identify large granules in specific subtypes of glial cells ("glial granules") of the adult *Drosophila* brain which contain polypeptides with previously characterized roles in germ cells including scaffold Tudor, Vasa, Polar granule component and Piwi family proteins. Interestingly, our super-resolution microscopy analysis shows that in the glial granules, these proteins form distinct partially overlapping clusters. Furthermore, we show that glial granule scaffold protein Tudor functions in silencing of transposable elements and in small regulatory piRNA biogenesis. Remarkably, our data indicate that the adult brain contains a small population of cells, which express both neuroblast and germ cell proteins. These distinct cells are evolutionarily conserved and expand during aging suggesting the existence of age-dependent signaling. Our work uncovers previously unknown glial granules and indicates the involvement of their components in the regulation of brain transcriptome.

[1] Department of Biological Sciences, Murray State University, Murray, KY 42071, USA. [2] Computer Science and Engineering Department, University of Louisville, Louisville, KY 40292, USA. ✉email: aarkov@murraystate.edu

 1

Germline (germ) cells give rise to every cell type in the next-generation organism after the union of the egg and sperm[1,2], and they express specific proteins and RNA that are assembled into large and dynamic membraneless granules referred to as germ granules[3–9]. Components of germ granules are required for germline fate[1,9,10]. In particular, Tudor (Tud) domain-containing proteins perform the molecular scaffold role recruiting other germ cell proteins into germ granules[3]. In fruit fly Drosophila, Tud protein is required for embryonic germ cell formation and, accordingly, the embryos laid by tud mutant mothers give rise to sterile adults (grandchildless phenotype)[11,12]. Tud protein contains eleven 50–55 amino acid Tud domains, which in germ granules, recognize methylated arginines of Piwi family proteins[1,13–15]. In different animals, Piwi family proteins and their associated small noncoding guide RNAs (Piwi-interacting RNAs, piRNAs) play central role in silencing of transposable elements in the gonad[16,17]. In addition, the founding member of this family, Drosophila Piwi protein, was shown to be autonomously required for stem cell maintenance in germline and some somatic stem cells[18,19].

Although a full-length tud cDNA was isolated from the fly's head in early cDNA collections, potential expression and somatic function of Tud scaffold, presumably, in the adult brain, remained unknown. Here we show that Tud and other germ cell proteins, previously studied for their unique roles in the germline, are expressed in the adult brain, and overall, these germ cell polypeptides assemble in large granules in glia (here referred to as glial granules), contribute to the genome integrity and regulate brain transcriptome. In addition to the localization of germ cell proteins to the glial granules, surprisingly, we identified a distinct population of cells in the adult brain that express both piwi and neural stem cell (neuroblast) marker gene, deadpan (dpn). Our study indicates the roles of genes in the adult brain whose functions have been traditionally viewed as being specific to germ cells.

## Results

**Germ cell gene Tud is expressed in glia in the brain.** To determine whether Tud is expressed in the brain, we prepared brain protein extracts and were able to detect Tud in the wild-type brain and not in tud protein-null mutant (tud[1], hereafter referred to as "tud mutant") with an anti-Tud antibody routinely used to identify Tud expression in the germline[20] (Supplementary Fig. 1d). However, we could not identify what specific brain cells express Tud in immunohistochemistry experiments, as this antibody gave rise to a high nonspecific background in the immunostained brains (Supplementary Fig. 1e). Therefore, to enable Tud detection in the brain, we tagged endogenous tud gene with N-terminal green fluorescent protein (GFP)- and FLAG-tags using CRISPR-Cas9 methodology, and used specific antibodies against the tags in immunostaining experiments (Supplementary Fig. 1a, b). Insertion of the N-terminal tags in tud locus did not affect primordial germ cell formation, expression or normal distribution of the protein in the germline (Supplementary Fig. 1c).

Immunostaining of the whole-mount brains with anti-GFP and anti-FLAG antibodies to detect Tud and different brain markers indicated that Tud is expressed in the brain glia labeled with antibody against Reversed polarity (Repo) protein, which is a specific marker for all types of glial cells (Fig. 1a).

As there are several subtypes of glial cells (Fig. 1), we next asked what specific glial cells express Tud. After crossing FLAG-tagged tud into genetic background of the fly lines that label different glial cells with membrane GFP-mCD8 marker, we determined that Tud is expressed in the cytoplasm of surface glia

(perineurial glia (PG) and subperineurial glia (SPG)), as well as cortex glia (CG) (Fig. 1d–h and Supplementary Fig. 2). In addition, super-resolution microscopy showed the assembly of Tud into the glial granules (Fig. 1c, g, h).

**Transposable elements are upregulated in tudor mutant adult brains.** In germline, Tud is required for the assembly of germ granules, which have been implicated in posttranscriptional regulation of germline gene expression, including silencing of transposable elements (retrotransposons)[13,21]. However, Tud role in the brain was not known. Here, using genome-wide RNA sequencing (RNA-seq) approach for tud mutant brains, we asked whether tud mutant affects the brain transcriptome. In addition, we compared the brain and ovarian transcriptomes of tud mutant flies. For these and all other experiments reported below, fly brains were carefully dissected by hand to exclude any cross-contamination from the germline-containing tissues (ovaries and testes). We found that 62 transposable elements were upregulated in tud mutant brains. Also, 21 transposable elements were upregulated in tud mutant ovaries (Fig. 2a and Supplementary Fig. 3), demonstrating that a Tud function in the brain is to silence transposable elements.

In the germline, transposable elements are silenced by Piwi proteins associated with small guide RNAs (piRNAs)[16,17]. Therefore, using small RNA deep-sequencing approach and rigorous filtering of piRNAs from piRNA database (piRBase)[22] to exclude RNAs mapping to tRNAs, rRNAs, snRNAs and snoRNAs, we identified 5590 piRNAs (found with a minimum count of 10 sequences for each piRNA) expressed in the wild-type brains. Using the same approach, we identified the 77,388 most abundant piRNAs in the wild-type fly ovaries.

piRNAs from the wild-type and tud mutant brains were compared, and 326 differentially expressed (DE) piRNAs were identified. Strikingly, the majority (85%) of the DE piRNAs were downregulated in tud mutant brains ($p < 0.01$) (Fig. 2b, Supplementary Fig. 4, and Supplementary Data 1).

Quite contrary to the detection of mostly downregulated piRNAs in tud mutant brains, tud mutant ovaries had somewhat less downregulated than upregulated piRNAs compared to the wild-type ovaries, 39% and 61%, respectively (13,099 ovarian DE piRNAs were identified) (Fig. 2c, Supplementary Fig. 4, and Supplementary Data 2), indicating that although Tud is involved in piRNA biogenesis in both brain and germ cells, there are distinct differences in how it regulates piRNA production in the brain and germline. Although piRNAs play a crucial role in silencing of transposable elements in germline[16,17], the observed differences between DE piRNAs in the brain and ovary in tud mutant might be expected based on the differences in development and function of these organs and suggest that, in addition to transposable elements, brain piRNAs may help to regulate brain non-transposon genes.

To explore the Tud-dependent mechanisms in piRNA production in the brain and ovary in more detail, we analyzed the population of piRNAs mapped to transposable elements in both tissues. In the wild-type brains, we detected more piRNAs mapped to sense strands of transposable elements than the antisense piRNAs (Fig. 2d), which was very different from piRNAs mapping to transposons in the wild-type ovary that showed the reverse (antisense) piRNA bias (Fig. 2e).

Next, we asked whether Tud plays a role in the maintenance of these transposon piRNA biases in the brain and ovary. Remarkably, in the tud mutant brains, the sense piRNA bias was eliminated demonstrating that Tud functions in the brain to maintain the bias toward transposon sense piRNAs (Discussion). Contrary to the brain piRNAs and similar to the wild-type

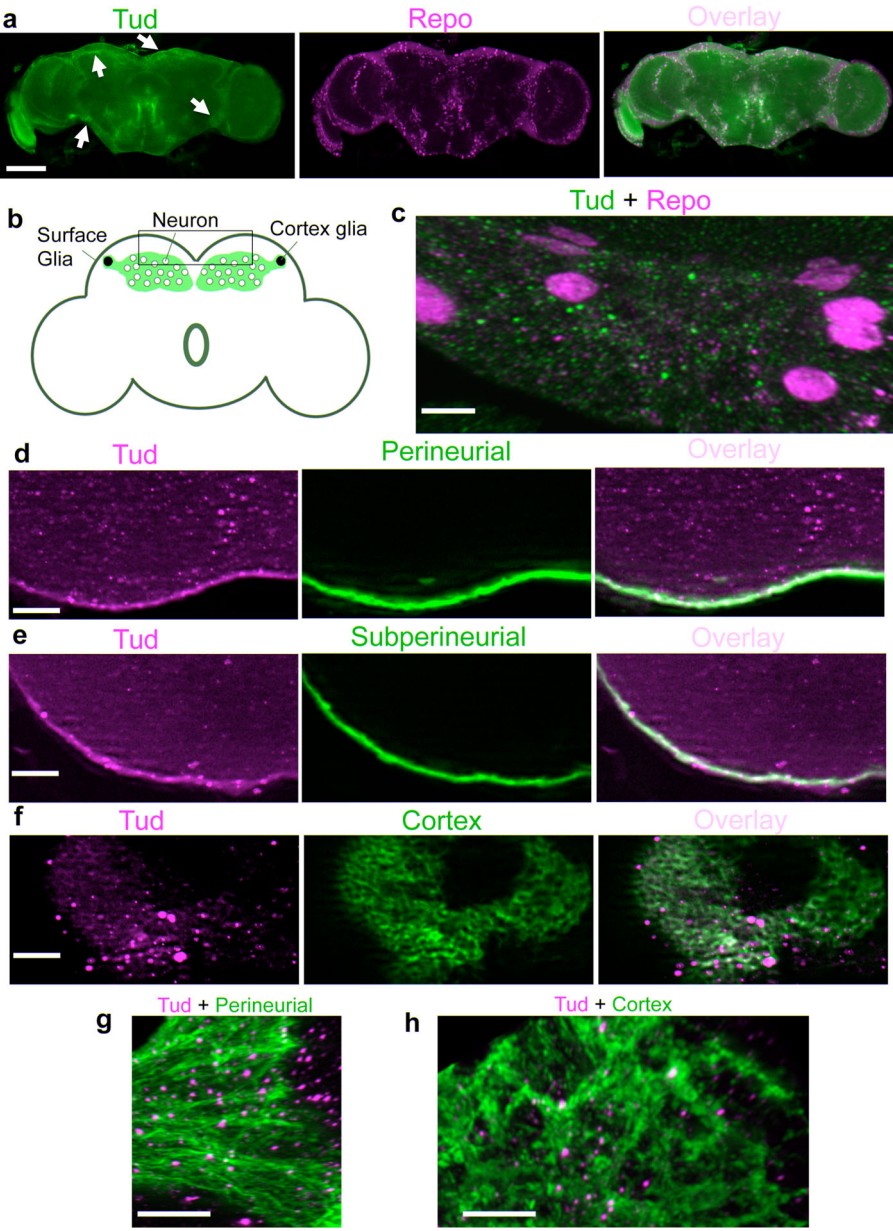

**Fig. 1 Tudor protein is expressed in perineurial, subperineurial, and cortex glia. a** GFP-tagged endogenous Tud (green channel) is expressed in glial cells (indicated with arrows, glia nuclei labeled with anti-Repo antibody, magenta) in the adult brains. **b** A diagram of the fly adult brain with surface and cortex glia subtypes indicated. A cortex glial cell envelops multiple neuronal bodies forming a honeycomb-like architecture. Nuclei of two cortex glial cells are shown in black. General area of the central brain imaged in **c–h** is indicated with a box. **c** High-magnification super-resolution optical section shows Tud localization (green) to glia (Repo-labeled, magenta). **d–f** Low-magnification optical sections show FLAG-tagged Tud localization (magenta) to perineurial (**d**), subperineurial (**e**), and cortex (**f**) glia. Different glia subtypes were labeled with the membrane marker GFP-mCD8 (green). **g, h** 3D high-magnification images of Tud granules (magenta) assembled in perineurial (**g**) and cortex (**h**) glia labeled with the membrane marker GFP-mCD8 (green). These images are composites of 186 optical sections (perineurial glia) and 225 sections (cortex glia) obtained with super-resolution confocal microscopy. Scale bars in **a** is 100 μm; in **c** is 5 μm; in **d–f** are 20 μm; and in **g, h** are 10 μm.

ovaries, *tud* mutant ovaries showed strong antisense transposon piRNA bias (Fig. 2d, e).

**Tud regulates brain transcriptome**. Comparison of piRNAs in the wild-type brain and ovaries highlighted the striking difference between these two piRNA populations. In particular, 10% of brain piRNAs were mapped to transposable elements and the majority of brain piRNAs was mapped to non-transposon genes (Fig. 2f). In a stark contrast, in the ovaries, about 90% of piRNAs were mapped to transposable elements and only about 3% of the piRNAs was mapped to the genes (Fig. 2f). This distribution of piRNAs was not

affected in *tud* mutant brains and ovaries. As the majority of brain piRNAs was mapped to the genes (genic piRNAs), we analyzed their distribution within the genes in more detail.

The majority of brain genic piRNAs were mapping to exons and 3′-untranslated regions (3′-UTRs) (62% and 22%, respectively, Fig. 2g). Contrary to that, most of the ovarian genic piRNAs were mapping to introns (66%), and piRNAs mapping to exons and 3′-UTR constituted the least abundant class of ovarian genic piRNAs (12 and 7% respectively). Genic piRNAs from *tud* mutant brain and ovaries showed similar distribution within the gene regions as their respective wild-type controls.

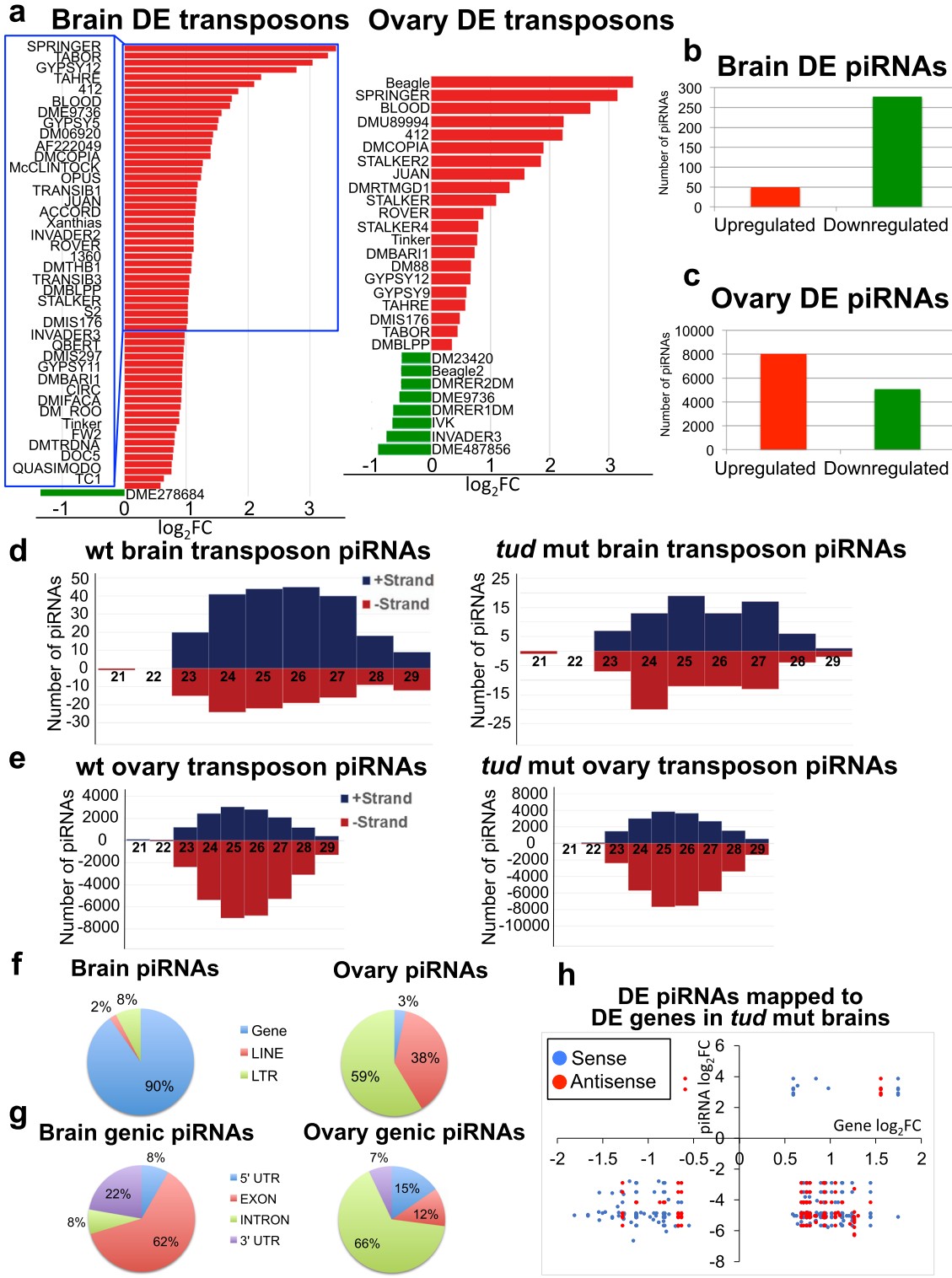

The relative abundance of genic piRNAs in the brain suggested that in addition to the transposable elements, non-transposon genes might be regulated by the piRNA-dependent mechanisms. Consistent with this hypothesis, similarly to the total brain piRNAs, ~90% piRNAs DE in *tud* mutant brains mapped to non-transposon genes with the majority of those mapping to exons (61%) and 3′-UTRs (19%) (Supplementary Fig. 4). Therefore, we asked whether Tud function in the brain may contribute to the regulation of the non-transposon transcriptome in the brain. In addition, given gender-specific behavioral aspects in flies, such as

courtship behavior, and thereby potentially different gender-specific outcomes of Tud-regulated gene expression in the brain, we dissected female and male brains separately and analyzed their transcriptomes and compared those with ovarian transcriptome in *tud* mutant background using RNA-seq approach.

We found that in *tud* mutant brains, there were 1749 DE genes (836 up- and 913 downregulated compared to the wild-type brains, false discovery rate (FDR) ≤ 0.05), which was substantially more than the number of genes DE in *tud* mutant ovaries (436 including 253 up- and 183 downregulated compared to the wild-

**Fig. 2 Transposable elements, non-transposon genes, and piRNAs are affected in *tud* mutant brains. a** Many transposable elements are upregulated in *tud* mutant brains (left panel) and, to a lesser extent, in the ovaries (right panel). For the brain, the names of transposable elements differentially expressed (DE) in *tud* mutants more than twofold are listed, and for the ovaries, all DE transposable elements are listed (complete list of DE transposons is included in Supplementary Fig. 3). *X*-axis shows $\log_2$ values of changes in the levels of transposable elements in *tud* mutants vs. wild-type (wt; fold change, FC). Although most transposable elements are upregulated (red bars), some transposons are downregulated (green bars) in *tud* mutants, $p < 0.05$ (source data are included in Supplementary Data 3). **b** In *tud* mutant brains, most piRNAs are downregulated. **c** In *tud* mutant ovaries, somewhat more piRNAs are upregulated than downregulated. **d** Tud functions in the brain to maintain sense piRNA bias. piRNAs were mapped to either "+" or "−" strand of transposons in wt (left panel), and *tud* mutant (right panel) brains. Lengths of piRNAs are indicated on *X*-axis and their corresponding numbers are shown on *Y*-axis. Wt brain piRNAs show sense (+ strand) bias, which is completely eliminated in *tud* mutants. **e** Wt ovarian piRNAs' antisense bias (left) is not affected in *tud* mutant ovaries (right). **f** Most of wt brain piRNAs are mapping to non-transposon genes, whereas the majority of wt ovarian piRNAs are mapping to transposable elements. Pie charts show percentage distribution of wt piRNAs mapped to non-transposon genes or LTR and LINE transposable elements. Also, small fraction of piRNAs was mapped to satellite DNA in the brain and ovary, 0.2% and 2.3% respectively. **g** wt brain piRNAs mapped to non-transposon genes (genic piRNAs) preferentially localize to exons and 3′-UTRs but in the ovaries, genic piRNAs preferentially localize to introns. **h** In *tud* mutant brains, most antisense piRNAs, which localize to DE genes, are downregulated and map to upregulated genes. Plots show $\log_2$FC values for DE genes (*X*-axis) vs. those for DE piRNAs (*Y*-axis) that map to these DE genes in *tud* mutant brains (source data are included in Supplementary Data 3).

type ovaries, FDR ≤ 0.05), indicating that Tud plays a more significant role in the regulation of non-transposon transcriptome in the brain than in the ovary. Importantly, genes related to translation, metabolism, and synaptic transmission were DE in *tud* mutant brains based on Gene Ontology and Kyoto Encyclopedia of Genes and Genomes statistically enriched terms (Supplementary Fig. 5). There were some notable differences between DE genes in female and male mutant brains. In particular, contrary to males, mutant females showed highly significant upregulation of *Neuropeptide-like precursor 4* (*nplp4*) and *target of brain insulin* (*tobi*) genes ($p = 5 \times 10^{-5}$). Neuropeptides and insulin signaling have been implicated in multiple aspects of sex-specific behavior and physiology including mating and reproduction[23,24], and, as our data indicate, brain-expressed Tud may partially contribute to these processes in *Drosophila*. Interestingly, a recent study also showed the upregulation of *tobi* in the fly heads in response to the accumulation of β-amyloid peptide (Aβ) in the Alzheimer's disease *Drosophila* model[25].

Further analysis of DE genic piRNAs, which map to DE genes in *tud* mutant brains, showed that the majority of downregulated antisense piRNAs map to upregulated genes (Fig. 2h), supporting the mechanism of gene regulation by antisense piRNAs, controlled by Tud.

**Germ-like glial granules**. Tud is assembled into cytoplasmic germ granules. Germ granules include nuage (around nurse cell nuclei in developing egg chamber) and polar granules (in the posterior of fly oocytes and embryos) in a specialized cytoplasm referred to as germ plasm[20]. Germ plasm is necessary and sufficient for the formation of primordial germ cells since they ectopically form in the opposite (anterior) pole after transplanting the germ plasm material there and polar granule components (Pgcs) are required for germ cell development[26,27]. In addition to Tud, germ granules include Vasa (Vas), Pgc, and Piwi family proteins such as Ago3 and Piwi. Interestingly, using super-resolution microscopy imaging, we detected polar granule-like Tud particles in glia (Fig. 1c, g, h) and, depending on a given protein, 18–49% of these glial Tud granules also contained Vas, Pgc, Ago3, and Piwi (141–287 glial granules were analyzed for each protein, Fig. 3, close-up super-resolution microscopy data are shown in Fig. 3e–i, Supplementary Figs. 6 and 7, and Supplementary Movies 1 and 2). To the best of our knowledge, this is the first evidence of polar granule-like structures in glia, which is consistent with their role in posttranscriptional control of gene expression and Tud function in transposon silencing, piRNA biogenesis and transcriptome regulation in the brain shown in this work.

In addition to being assembled in glial granules, unexpectedly, Ago3 was frequently found in the cytoplasm of some neurons clustered together in CG forming the "honeycomb"-like architecture (Fig. 3f, g and Supplementary Movies 3 and 4), suggesting a previously unknown role of Ago3 in the cell bodies of a subset of neurons. Importantly, although Tud and Ago3 are found in the same granules in glia (Fig. 3f, Supplementary Fig. 7b, and Supplementary Movie 1), Tud was never detected in the Ago3-positive neuronal bodies embedded in the CG, consistent with glia-specific expression of Tud (Fig. 3f, g).

**Adult brain contains Piwi- and Dpn-expressing cells, which are embedded in CG and expand during aging**. From the detailed imaging experiments described above, which showed Tud and Piwi in glial granules, we also noticed a very small number of specific cells that expressed Piwi but not Tud in the upper middle part of the adult brain (Fig. 4a). These Piwi-positive cells are usually found symmetrically located on dorsal side in both halves of the middle brain at a distance of ~150 μm from each other and Piwi is exclusively cytoplasmic. As Piwi has been shown to be autonomously required for stem cell maintenance in both germline and soma[18,19], we hypothesized that these cells might be a resident stem cell population in the adult brain. Therefore, we co-stained these cells with Piwi and a pan-neuroblast-specific marker Dpn, and, consistent with our hypothesis, we found that both Piwi and Dpn specifically colocalize in these cells (Fig. 4b). Although Dpn is a nuclear protein in neuroblasts, in these Piwi-positive cells Dpn is localized to the cytoplasm. The finding of the Piwi/Dpn-expressing cells in the adult brain was unexpected. Furthermore, we determined that genomic GFP-tagged Piwi transgene[28] is also expressed in these specific brain cells and colocalized with Dpn, thereby providing another line of evidence for our identification of the Piwi/Dpn-expressing cells in the brain (Supplementary Fig. 8).

Next, we analyzed the location of these Piwi/Dpn-positive (+) cells in more detail. In particular, we tested where these Piwi/Dpn+ cells are situated in relation to glia. We found that these cells are embedded in CG as shown with the CG-specific membrane marker GFP-mCD8 and anti-Piwi/-Dpn antibodies co-labeling experiments (Fig. 4c, d).

What may be a role of these Piwi/Dpn+ cells in the adult brain? We proposed that these cells might be required to replenish the population of non-functional neurons and glial cells during normal aging. If true, then one would expect to see an increase in the number of these cells during aging as a response to the need to replace the aging brain cells. Therefore, we immunostained brains at different ages (from 1, 3, 7, and 32-day-old flies) with anti-Piwi and anti-Dpn antibodies and found a substantial increase in the number

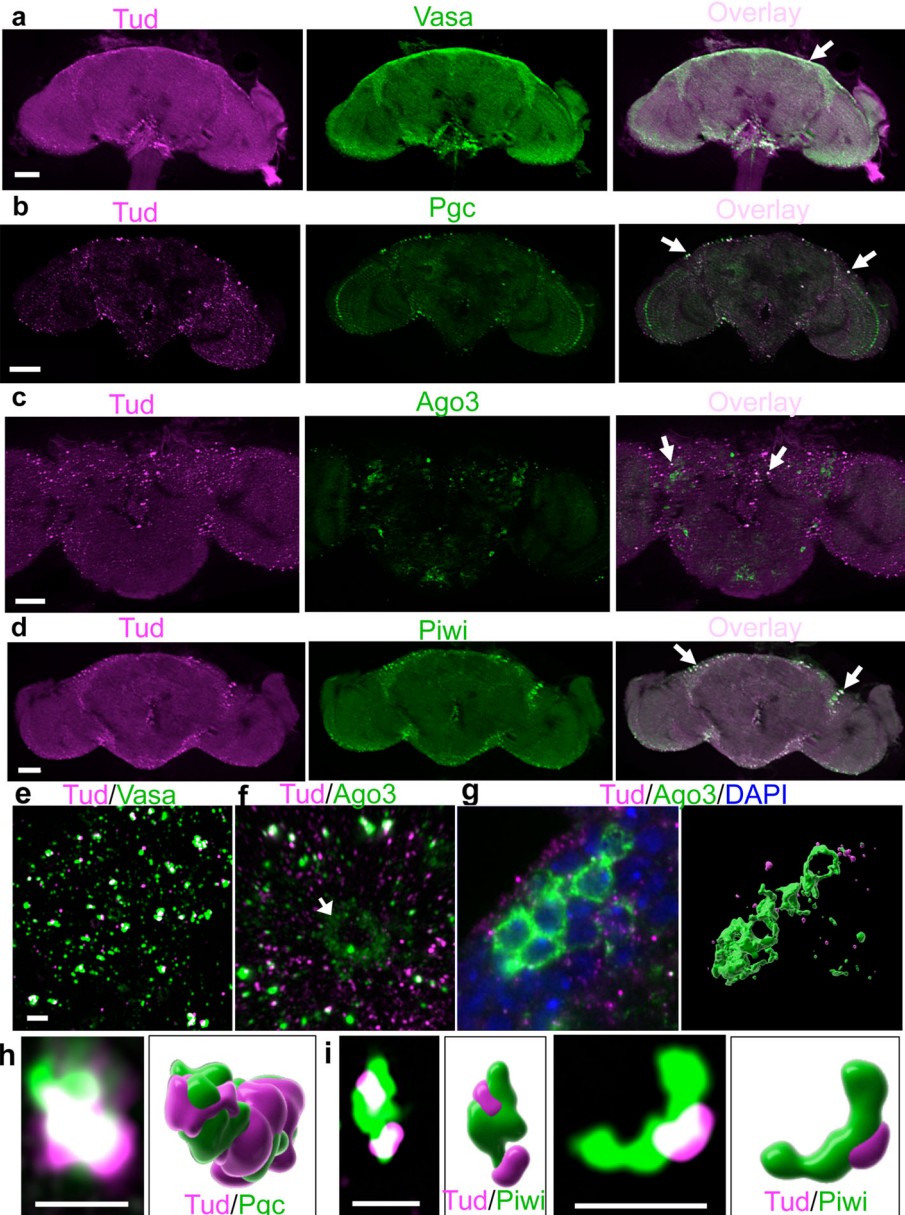

**Fig. 3 Vasa, Polar granule component, Ago3, and Piwi colocalize with Tudor in glial granules. a–d** Optical sections of the adult brains immunostained with anti-FLAG antibody to label Tud (magenta) and Vasa (**a**), Pgc (**b**), Ago3 (**c**), and Piwi (**d**) (green) show the localization of all these proteins in glia. Arrows point to colocalized foci. **e, f** Super-resolution images with Tud and Vas (**e**) and Ago3 (**f**, Supplementary Movie 1) glial granules. In addition to glia, Ago3 is expressed in neurons which do not express Tud (a neuronal cell body is indicated with an arrow (**f**)). **g** In neurons, Ago3 is frequently expressed in the cytoplasm of several neuronal cell bodies clustered in the cortex glia (green) with a characteristic "honeycomb" appearance (super-resolution optical section, left panel). Right panel shows 3D reconstruction of the Ago3-positive neurons (green) and Tud glial granules (magenta) corresponding to the left image. **h, i** Super-resolution optical sections and corresponding 3D reconstructions of Tud/Pgc (**h**, Supplementary Movie 2) and Tud/Piwi (**i**) individual glial granules (Tud and Pgc/Piwi are labeled with magenta and green respectively). Scale bar in **a**, **b**, **d**, and **c** is 50 μm and 40 μm, respectively; 2 μm scale bar in **e** is the same for **f** and **g**. Scale bar in **h** and **i** is 2 μm.

of Piwi/Dpn+ cells during aging. In particular, the majority of 1-day-old brains show only 2 Piwi/Dpn+ cells (9 out of 13 brains counted), whereas the majority of the 32-day-old brains have 4–9 cells (6 out of 10 brains counted) (Fig. 4e, f).

Next, we explored whether the presence of these Piwi/Dpn+ cells is evolutionarily conserved. To this end, we immunostained brains from other *Drosophila* species, which have been diverging from their common ancestor for estimated 25–55 million years[29,30] (*Drosophila simulans*, *Drosophila yakuba*, and *Drosophila pseudoobscura*), with anti-Piwi and anti-Dpn antibodies. Similar to *D. melanogaster*, in all the other species, the Piwi/Dpn

+ cells were found (Supplementary Fig. 9). Interestingly, in *D. yakuba*, these cells are observed not only on the dorsal side as seen in other *Drosophila* species but also there are additional cells that are ventrally located (Supplementary Fig. 9b).

**Piwi/Dpn-expressing cells form long extensions in adult brains.** Remarkably, we found that long, thin (~1 μm in diameter) Piwi/Dpn+ extensions emanate from Piwi/Dpn+ cells located on the opposite sides of the brain described above (Fig. 5 and Supplementary Movie 5). The extensions converge into the brain midline and either end there (Fig. 5 and Supplementary Movies 5

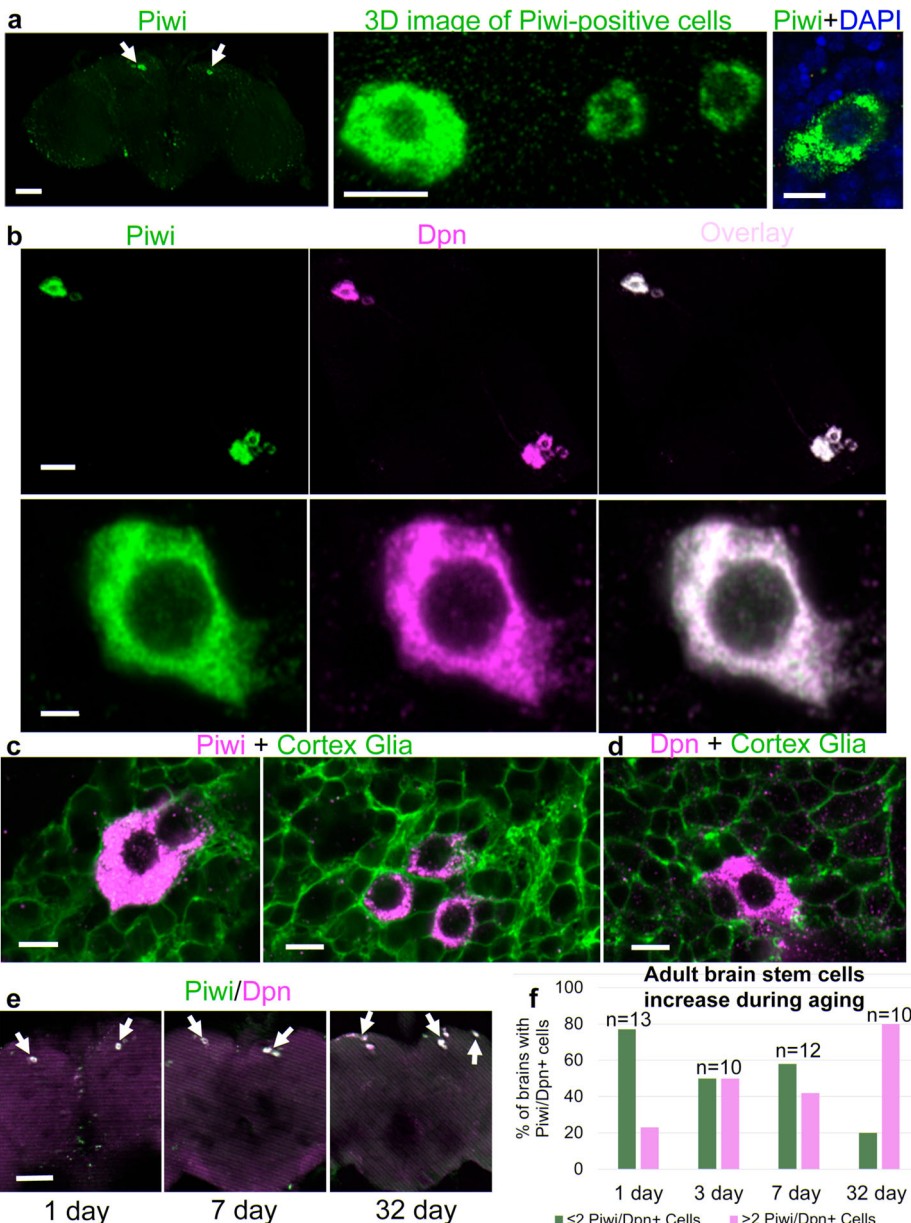

**Fig. 4 Population of Piwi/Dpn-expressing cells in the cortex glia of the adult brain expands during aging. a** Left panel, an optical section of the whole brain showing symmetrically located Piwi-expressing cells (arrows). Frequently, Piwi-positive cells are heterogeneous in size and Piwi-expression levels and clustered together in each half of the brain (middle panel). Piwi is exclusively cytoplasmic (right panel). **b** Co-staining of brains with anti-Piwi (green channel) and anti-Dpn (magenta) antibodies shows colocalization of both proteins in the cells. The images are optical sections obtained with the super-resolution confocal microscopy. The top panels for each channel were generated from six tiles, individually imaged with super-resolution microscopy, to show all cells in a single brain. The bottom panels show super-resolution microscopy images of Piwi and Dpn colocalizing in a single cell. **c**, **d** Piwi (**c**) and Dpn (**d**)-positive cells (magenta) are enveloped by cortex glia labeled with membrane marker GFP-mCD8 (green). **e** The number of Piwi-(green) and Dpn-(magenta)-expressing cells (indicated with arrows) is increased in aging brains (overlay images of brains at a specified age are shown). **f** Percentage of brains with Piwi/Dpn-positive cells with two or less cells (green bars) and more than two cells (magenta bars) are shown for four different ages. Number of brains counted for each age is indicated above the bars. Scale bar in **a** (left panel) is 50 μm, **a** (middle panel) 10 μm, **a** (right panel) 5 μm, **b** (top panels) 20 μm, **b** (bottom panels) 2 μm, **c** and **d** 5 μm, and **e** 50 μm.

and 6) or, in some cases, turn in different directions along the midline (Supplementary Movie 7). In the midline, there is a cell or cell cluster that weakly expresses Piwi but not Dpn (Fig. 5a, c and Supplementary Movies 5 and 6). These extensions can also be seen in *D. simulans*, indicating their evolutionary conservation and functional importance (Supplementary Movie 8). Due to the fact that these cells are very far apart, the extensions may provide a mechanism for coordinated response by the Piwi/Dpn+ cells to

a yet unknown signal (Fig. 6), which may trigger their proliferation. This idea is consistent with our data showing that the number of Piwi/Dpn+ cells is increasing during aging (Fig. 4) implying that the signal production may also be age-dependent. Alternatively, these Piwi/Dpn+ cells may produce a signal themselves and use these projections to deliver signal molecules similarly to the transport in the axonal projections of neurosecretory cells[24] (Discussion).

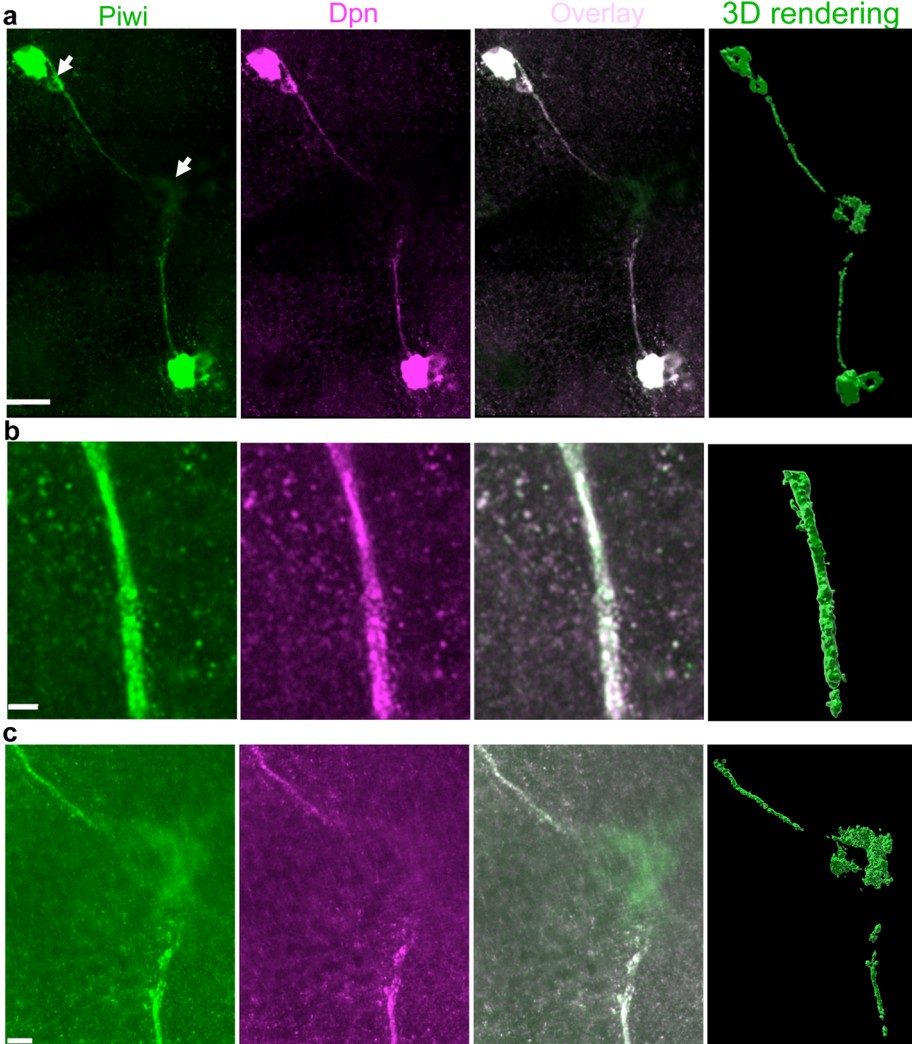

**Fig. 5 Piwi/Deadpan-expressing cells form long extensions in adult brains. a** Super-resolution imaging of Piwi (green channel) and Dpn (magenta)-positive extensions, which emanate from the cells and end in the brain midline converging to a Piwi-positive cell (indicated with arrow at the middle of the left panel). Also, a smaller Piwi/Dpn+ cell appears to be connected to a larger neighboring cell by the extension (indicated with top arrow). **b** A super-resolution optical section showing details of an extension's segment. **c** A super-resolution optical section of the brain midline showing details of the central segments of the extensions converging in the Piwi-positive cell. Right panels in **a**–**c** show the corresponding 3D reconstructions based on super-resolution optical sections (green channel). To visualize extensions, the images were generated with either high laser power during acquisition or post-acquisition by increasing the signal intensity with Imaris software. Scale bars in **a**, **b**, and **c** are 20, 2, and 5 μm respectively.

## Discussion

In this study, using super-resolution microscopy, we identified, to the best of our knowledge, new structures in glia of the *Drosophila* adult brain ("glial granules"), which contain germ granule proteins required for germline development including Tud, Vas, Pgc, Ago3, and Piwi. In addition, we show that in the adult brain, there is a small population of cells in two symmetrical locations of the central brain, which express neuroblast and germ cell proteins Dpn and Piwi, respectively, and these cells expand during aging. Furthermore, these Piwi/Dpn+ cells are evolutionarily conserved, reside in the CG and, remarkably, they show long (50–100 μm) extensions, which converge in the midline of the brain at the Piwi-specific cell cluster.

The presence of multiple germ granule components in glial granules was unexpected and suggested that, similar to germ cells, adult brains may employ glial granule components for post-transcriptional gene regulation mechanisms (Fig. 6). Accordingly, using next-generation sequencing approaches, we demonstrated that central molecular scaffold Tud protein functions in the brain

to silence transposable elements and also regulate the non-transposon transcriptome. Our work showed that Tud is also required for biogenesis of piRNAs in the brain and suggests that antisense piRNAs, downregulated in the absence of Tud, target brain genes to regulate their products' levels.

We demonstrated that piRNA profiles in the fly brain and ovary are different and Tud protein has important distinct roles in piRNA biogenesis and is required for the silencing of transposable elements in both organs. Interestingly, contrary to the ovary, in the brain, Tud functions to maintain a transposon sense piRNA bias which is lost in the *tud*-null mutant brains (Fig. 2d). How does Tud maintain a sense piRNA bias in the brain? Given the role of Tud in germ cells as a molecular scaffold[11,31], this protein may be required for the assembly of efficient transposon silencing complexes which cleave transposable elements thereby producing abundant transposon sense piRNAs. Consistent with this mechanism, we demonstrated in this study that in the brain Tud is assembled with important piRNA biogenesis factors, including Piwi family proteins and Vas RNA helicase, in glial

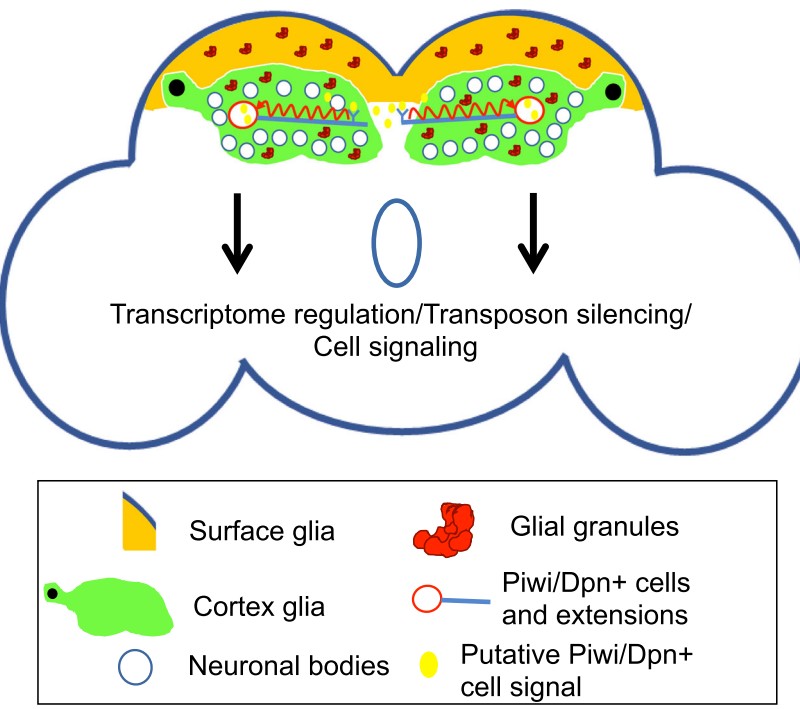

**Fig. 6 Summary schematic and model.** Glial granules assembled from Tudor protein scaffold and other germ cell proteins, including Vasa, Polar granule component, and Piwi, function in brain transcriptome regulation. Evolutionarily conserved Piwi/Dpn-positive cells have been identified in the adult *Drosophila* brain and show remarkably long extensions which may be involved in signal transmission caused by aging, memory formation, or perturbations in the brain milieu such as pathogen invasion.

granules, and that the lack of Tud leads to the upregulation of many transposable elements. However, we cannot rule out a less direct role of Tud in the maintenance of the sense piRNA bias in the brain, which may be mediated through another factor whose expression level is regulated by Tud. In particular, similar to piRNAs detected in the wild-type brain, although 13% of piRNAs DE in *tud* mutant brains map to transposable elements, majority of these DE piRNAs are genic piRNAs and most of them map to exons and 3′-UTRs of non-transposon genes (Supplementary Fig. 4a, b). Contrary to the brain DE piRNAs, in the ovary, most DE piRNAs map to transposable elements and only 4% are the genic piRNAs with the majority of those mapping to introns. The predominance of genic piRNAs in the brain suggests a role of piRNAs in the regulation of brain transcriptome and the role of *tud* in this process. Consistent with this idea, we found that most of the antisense piRNAs that are downregulated in *tud* mutant brains map to the genes that are upregulated in the mutant brains (Fig. 2h) indicating a Tud-controlled mechanism of antisense piRNA-directed regulation of brain transcriptome, which, in the absence of Tud, leads to the upregulation of certain genes due to the decrease in the levels of the corresponding antisense piRNAs. Furthermore, in support of this mechanism, we demonstrated that the lack of Tud in the brain mostly leads to the downregulation of piRNAs (85% of total DE piRNAs) and the levels of downregulated piRNAs are reduced much more than the increase in the levels of the upregulated piRNAs in the mutant brains (Supplementary Fig. 4c). Therefore, these data indicate that Tud has an important role in the production of piRNAs in the brain and suggest that the role of these piRNAs is to prevent gene overexpression in the brain.

In the germline, sense and antisense piRNAs are amplified by the Ping-Pong cycle mechanism, whereby antisense piRNA guides a specific Piwi protein to cleave a transposon RNA between its nucleotides corresponding to the nucleotides 10 and 11 of the antisense piRNA, which results in the production of

transposon-derived sense piRNA. Subsequently, this sense piRNA guides a cleavage of the antisense piRNA precursor to generate an antisense piRNA. Therefore, the first ten nucleotides of the transposon-derived sense piRNAs and the corresponding antisense piRNAs are complementary. We could not detect the Ping-Pong preference for 10 nt overlap in the brains piRNAs from either wild-type or *tud* mutant brain demonstrating the lack of the efficient Ping-Pong mechanism for piRNA production in the brain. Contrary to that, the Ping-Pong cycle was similarly active in both wild-type and *tud* mutant ovaries (The Ping-Pong z-score, which is a relative value calculated to measure the preference of the 10 nt complementarity between sense and antisense piRNAs[32,33], was 8.6 and 11.2 in wild-type and *tud* mutant ovaries, respectively), indicating that Tud does not participate in the Ping-Pong cycle in the germ cells.

The differences in the piRNA populations in brain and ovary and the distinct aspects of Tud function in the piRNA biogenesis in brain and germ cells uncovered by this work indicate that both organs use Tud scaffold and its associated proteins to regulate distinct classes of piRNAs and brain- and germline-specific transcriptomes.

The presence of cells in the adult brain that express both Piwi and Dpn and show long Piwi/Dpn+ extensions is intriguing. What role might these cells and the extensions play? Although Dpn is a neural stem cell marker and Piwi is intrinsically required for stem cell maintenance in germline and soma[18,19], we cannot rule out that these Piwi/Dpn+ cells are differentiated cells, with, however, unexpected and not previously described co-expression of Dpn and Piwi. Interestingly, long protrusions such as cytonemes and tunneling nanotubes detected in other cell types were implicated in intercellular signaling[34]. Similarly, the Piwi/Dpn extensions in the central brain may be involved in signaling that coordinate the response of the Piwi/Dpn+ cells to a specific signal. In our model, the signal may be produced in the midline region of the central brain in the proximity of both extensions,

which emanate from Piwi/Dpn+ cells on the opposite sides of the brain (Fig. 6). Furthermore, the signal may be secreted in this area due to an environmental condition such as pathogen (bacterial or viral) invasion or an environmental stimulus such as learning experience that results in the formation of new memory. Consistent with this model, these environmental stimuli could result in the generalized cell response potentially leading to the proliferation of Piwi/Dpn+ cells. Alternatively, signal molecules may be produced by Piwi/Dpn+ cells and then transported via the projections similarly to neurosecretory cells and their axonal projections[24]. Further research will be needed to determine whether these unusual Piwi/Dpn+ cells are in fact bona fide stem cells or function as differentiated cells.

Glial granules, described in this work, show distinct clusters of their protein components whose roles were mainly restricted to the development of germ cells where they assemble in membraneless RNA-protein granules (germ granules). Recently, it was shown that germ granules in *Drosophila* germ plasm are assembled from distinct partially overlapping protein and RNA homotypic clusters[5,7,8]. Similar to germ granules, glial granules show distinct protein clusters, which partially overlap, suggesting that the mechanisms of the assembly of glial granules is similar to germ granules. The detailed composition of glial granules will need to be determined to characterize the role of the granules in the brain; however, our current data suggest a remarkable conservation of the assembly mechanisms of large dynamic membraneless organelles in brain and germline.

## Methods

**tud CRISPR gRNA target selection and donor plasmid cloning.** CRISPR/Cas9 methodology was used to tag endogenous *tud* gene with GFP and FLAG-tags. These methods were detailed previously[35]. In particular, genomic DNA from Vas-Cas9 flies used for *tud* tagging (Bloomington Drosophila Stock Center, BDSC stock #55821) was isolated and a 3 kb fragment of the *tud* gene was PCR-amplified and sequenced to verify that there are no SNPs in the region of the *tud* gene where gRNA would target the *tud* locus encoding the N terminus of Tud protein. The donor plasmids were ScarlessDsRed vectors designed for efficient screening of tag insertions into a gene locus using DsRed fluorescence in the compound eyes and the ocelli. Specifically, the vectors have a cassette containing 3× P3 (eye-specific promoter) and DsRed flanked by two inverted repeats TTAA that can be recognized by the PBac (piggy bac) transposase. The flies that contained a given tag insertion expressed DsRed in the eyes. Subsequently, the cassette was removed from *tud* locus by crossing into pBac Transposase background (BDSC stock #8285). The fact that the cassette disrupts *tud* was used to confirm that homozygous females (*tud*^tag DsRed/*tud*^tag DsRed) show the *tud* mutant (grandchildless) phenotype. The subsequent removal of cassette with pBac Transposase rescued the *tud* grandchildless phenotype and the expression of tagged Tud in the ovary was then confirmed with western blotting and immunostaining/live-imaging experiments (Supplementary Fig. 1).

**Immunohistochemistry.** Whole-mount brain immunostaining experiments were described[36]. To identify specific glial subtypes that express Tud, endogenous FLAG-Tud was crossed into UAS-mCD8-GFP background where mCD8-GFP (BDSC stock #32184) was generated by Gal4 drivers specifically expressed in different glial subtypes as follows. For the PG, SPG, and CG, Gal4-*shn* (BDSC stock #40436), Gal4-*mdr65* (BDSC stock #50472), and Gal4-*wrapper* (BDSC stock #45784)[37] were used, respectively.

**Antibodies.** For brain immunostaining, tagged Tud was labeled with either rabbit anti-GFP (Abcam, 1:5000), mouse anti-FLAG (Sigma, 1:5000) or rabbit anti-FLAG (Novus Biologicals, 1:4000) antibodies. Other antibodies include mouse anti-Repo (DSHB, 1:200), guinea pig anti-Repo[38] (1:2500), mouse anti-Wrapper (DSHB, 1:400), rabbit anti-Vas[39] (1:2000), rabbit anti-Pgc[40] (1:2000), rabbit anti-Ago3[41] (1:1500), rabbit anti-Piwi (Sdix, 1:1500, Supplementary Fig. 8); and guinea pig anti-Dpn[42] (1:2400).

**Super-resolution microscopy.** Super-resolution microscopy was performed with Zeiss LSM 880/super-resolution Airyscan module system with inverted laser scanning confocal microscope AxioObserver and Plan-Apochromat ×63/1.4 Oil DIC M27 objective at Vanderbilt University Cell Imaging Shared Resource (CISR). Images analysis and 3D reconstructions were carried out using Imaris software (version 9.5, Oxford Instruments) and HP Z8 workstation. To measure the percentage of Tud glial granules that overlap with a given protein, Surfaces option in Imaris was used to automatically identify granules and calculate the percentage of particles which contain both proteins. Glial granules (141–287) were analyzed for each Tud/given protein pair.

**RNA-seq and piRNA sequencing.** For RNA-seq and piRNA sequencing, brains and ovaries from young *tud* protein-null flies (*y, w; tud*[1]*, bw, sp/Df(2R)Pu*^rP133)[11] and control, *tud* heterozygous flies (*y, w; tud*[1]*, bw, sp/CyO*, P[*w*+, *hs-hid*]) were dissected. Standard cornmeal-molasses medium was used for growing the flies at 25 °C. Great care was taken to ensure that the brain samples did not have contaminating tissues from other parts of the animal. Specifically, the anesthetized flies were transferred to a Petri dish where the head was removed and the body was discarded prior to brain dissection. Brains were dissected from the head capsule and transferred into a centrifuge tube with 1× phosphate-buffered saline (PBS). Fifteen to 30 brains were collected per dissection session, to minimize degradation of tissue. In the end of each session, PBS was removed from the tube and the brains were resuspended in 1 ml of fresh 1× PBS to remove any possible contaminants. Finally, all liquid was removed and the brains were frozen in liquid nitrogen and stored at −80 °C.

For small RNA-seq, RNA isolation was done using the mirVana miRNA isolation kit. Approximately 150 brains per biological replicate were used to yield the 1 μg of total RNA required for sequencing. Number of biological replicates for *tud* mutant brains and corresponding control was 2 and 3, respectively, generating data from four sequencing samples per replicate. The general construction of small RNA libraries was described previously[43,44]. Accordingly, *Drosophila* 2S rRNA was depleted from the total RNA samples and the sequencing libraries were constructed with TruSeq Small RNA kit (Illumina Catalog number RS-200-0012) in the presence of 25% (weight/volume) of Polyethylene Glycol 8000 in 3′-adapter ligation buffer. The sequencing was carried out with Illumina NextSeq 500 using the NextSeq 500/550 75 cycle High Output Kit v2 (FC-404-2005). Small RNA-seq from the ovarian samples (RNA from 40–50 μl of ovaries per biological replicate) was done similarly to the brain samples. In particular, three biological replicates each for *tud* mutant and control were obtained generating data from four sequencing samples per replicate.

For RNA-seq transcriptome analysis, *tud* mutant and control brains were dissected as described for small RNA-seq samples. In addition, to test for gender-specific aspects of transcriptome regulation by Tud, female and male brains were dissected separately and transcriptome analysis was done for each gender. Libraries were prepared using the TruSeq Stranded Total RNA LT Sample Prep Kit-Set A (Cat # RS-122-2301) with Ribo-Zero Gold with following modification. rRNA depleted RNA samples were made up to a total volume of 100 μl with Ultra Pure water. A size selection was performed to remove 5S rRNA and tRNA using the Zymo Research RNA Clean and Concentrator-5 kit (catalog number R1015) as described[45]. RNA libraries from samples for *tud* mutant and control ovaries were generated as for the brain samples. Three biological replicates were sequenced per each of the six conditions: (1) *tud* mutant female brains; (2) *tud* mutant male brains; (3) control female brains; (4) control male brains; (5) *tud* mutant ovaries; (6) control ovaries. For each biological replicate, sequencing data from two sequencing runs and four sequencing lanes per each run were generated. Sequencing was performed on Illumina NextSeq 500 using the NextSeq 500/550 150 cycle High Output Kit v2.

**Bioinformatics analysis.** Small RNAs that correspond to piRNAs from piRBase database[22] were filtered to remove sequences corresponding to tRNAs, rRNAs, snRNAs, and snoRNAs. Also, the sequencing data were filtered to identify the most abundant unique piRNAs which have a minimum count of ten sequences found across the samples.

The filtered minimum count list for *tud* mutant and control for the brain and ovary were then compared to determine differential expression using edgeR[46]. DE piRNAs in the brain and ovary were determined with *p*-value < 0.01 and FDR < 0.05 for the brain and ovary, respectively.

Gene features (including 5′-UTR, 3′-UTR, exon, and intron) were extracted from the dm3 FlyBase annotations[47] using annotations extracted from the UCSC genome browser[48]. These locations were compared to piRNA locations mapped to dm3 in piRBase using the custom script *findOverlappingFeatures.R*.

The ping-pong Z-scores were calculated using a PPmeter software[32].

DE transposon and non-transposon genes in *tud* mutant brains and ovaries, compared to corresponding controls, were identified from RNA-seq data. The raw fastq reads were mapped to the dm6 assembly using tophat2 (v2.0.3)[49] guided by the BDGP6 gtf file (v90). DE genes were determined using cuffdiff (v2.2.1)[50] based on an FDR cutoff of 0.05. A total of 179 transposons were downloaded from the FlyBase FTP and used for the analysis.

**Reporting summary.** Further information on research design is available in the Nature Research Reporting Summary linked to this article.

## Data availability

Data that support the findings of this study have been deposited in GEO as a superseries with accession GSE149750. All relevant data are available from the corresponding author upon request.

## Code availability
All custom scripts used in the analysis are available upon request.

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

## Acknowledgements
We thank A. Aravin, B. Altenhein, G. Hannon, J. Knoblich, R. Lehmann, A. Nakamura, Bloomington Drosophila Stock Center, and the National Drosophila Species Stock Center for the antibodies and fly lines. Also, we thank J. Schafer and the Vanderbilt Cell Imaging Shared Resource for help with Zeiss LSM 880/super-resolution microscopy, and S. Waigel and W. Zacharias from the University of Louisville Genomics Facility for next-generation sequencing. In addition, we thank Margaret Hagen for helping with some parts of the work. This work was supported by Kentucky Science and Technology Corporation grant KSEF-148-502-17-404 and National Science Foundation grant awards MCB-1715541 to A.L.A. Also, some parts of the work, including next-generation sequencing and bioinformatics analysis were funded by a grant from the NIH National Institute of General Medical Sciences, P20GM103436.

## Author contributions
A.L.A. and S.J.T. designed the experiments, and S.J.T performed them. E.C.R. carried out bioinformatics analysis. A.L.A. and S.J.T. interpreted the data and wrote the manuscript.

## Competing interests

The authors declare no competing interests.
