## [Peer Review File · Communications Biology]

Reviewers' comments:

Reviewer #1 (Remarks to the Author):

This manuscript presents novel data about germline proteins in the brain. On the one hand, the presence of granules of Tudor plus other germline proteins in glia and the role piRNAs regulating mostly non-transposon genes in this tissue. On the other hand, they describe a group of cells that express piwi and a neuroblast marker and have a very intriguing behavior and morphology.

The work and findings for the role of Tudor in transposable elements repression and piRNA biogenesis are solid and very interesting. Specially the fact that the majority of brain piRNAs map to genes in the brain. They are sufficient for a short article and a good start for further investigations. In this case a discussion similar to the supplementary discussion would fit better.

On the other hand, I find the evidence for calling the new and intriguing cell population identified "stem cell" weak. This statement is based on two facts: 1. the fact that these cells express a neuroblast marker and piwi but no extensive molecular characterization has been carried out, i.e. other neuroblast or neuronal markers; 2. the number of cells increase with age. While this might be a property of certain stem cells, a strong evidence for calling these cells stem cells would be lineage tracing and seeing postmitotic differentiated cells derived from them. More work in this part could lead to an additional paper in the future.

One way to maintain the two parts (Tudor results and new cell population) in the same paper would be to remove the term stem cell from the title, and rewrite intro and discussion accordingly. It is acceptable to hypothesize that they could be stem cells, as they do, but there is not enough evidence for using the term stem cell in the title. It would be misleading.

Specific comments/questions on the data.

1. Expression analysis of proteins in different glial types can be done with regular microscopy, is there any particular reason for using super-resolution in Figure 1?
2. What brain regions are the ones shown in Fig 1b to h? The images lack spatial context. It would be good to have a schematic showing which part of the brain is being imaged and its orientation.
3. The figure legend does not explain the two panels Fig1b. They look different, why? In the first Tud signal is specks and in the second Tud labels membranes? It is hard to relate this pattern to glia using a glial nuclear marker in this panel, the dots and membrane signal could be in neurons. Material and methods indicate that detection of signal is done with antibodies. It would be informative to see a secondary control given that, since this a genomic reporter, lack of signal in the mutant background control cannot be performed.
4. In the image provided for panel 1e it is difficult to see the spatial relationship between the red and green signal. Also it seems that the Tud-GFP (Fig 1b) gives a different pattern than Tud-Flag (Fig 1e).
5. And it is unclear what point wants to be made with panels Fig 1g and 1h. With the current image size it is hard to see well.
6. Are the Tud expression experiments in Fig 1 and Fig 3 done with homozygous Tud-tagged animals?
7. Have you noticed any behavioral defects in Tud mutant animals, given the presence of a significant number of DE expressed non-transposon genes? Maybe future experiments could include testing Tud mutants for different behavioral assays (<https://www.janelia.org/project-team/fly-olympiad>) in collaboration with Janelia Research Campus.

Reporting summary feedback: report is in accordance with the information provided in the manuscript

Reviewer #2 (Remarks to the Author):

Early cDNA collections isolated from the fly's head demonstrated the presence of tud cDNA, which encodes the Tudor (Tud) scaffold protein that has been best characterized in germ granule RNPs. In this study, Tindell, Rouchka, and Arkov validate the expression of Tud in the adult fly brain and show it is specifically expressed in the cytoplasm of perineurial, subperineurial, and cortex glia.

Surprisingly, they find that Tud coexpresses and colocalizes there with Pgc, Ago3, and Piwi, three other proteins with primary roles in the germline, to form what they call "glial granules." In the germline, germ granule components suppress transposable elements associated with piRNAs. They show that this is also the case in tud mutant brains; although, a sense-transposon piRNA bias is observed instead of the antisense transposon bias in tud mutant ovaries. piRNA sequencing in the brain also revealed that most map to non-transposon genes, unlike the ovaries, and that this coincides with more differentially expressed genes in tud brains than in tud mutant ovaries. Together, these findings implicate a post-transcriptional role for germ granule associated proteins in glia.

The author's investigations also reveal Piwi expression in a very small number of cells that coexpress the pan-neuroblast-specific marker Deadpan (Dpn). This novel class of cells is embedded in cortex glia and send extensions into the brain midline, and are conserved in other *Drosophila* species. These cells appear to proliferate in fully developed animals, suggesting they represent a stem cell population that replenishes non-functional neurons and glial cells in aging animals. These findings have broad implications for a wide audience, and further support proposed roles for germ granule proteins being repurposed to promote cellular plasticity during development and regeneration. On the whole, the paper is well written and succinct. The findings and interpretations are clear and supported by the data, and make a very significant contribution to our understanding of small RNA regulation and neuronal stem cells.

Minor comments follow.

- 1) On line 58, Tud expression is examined in the brain with anti-Tud antibody. It would help if images of this staining and the westerns could be included in extended Figure 1.
- 2) The paragraph beginning on line 121 that discusses the difference in piRNA biogenesis between the germline and brain could be expanded with references and something that addresses why this difference may or may not be expected. Then in the following paragraph, include an interpretation of the sense/antisense bias.
- 3) Line 167 references the supplementary discussion. This supplement is very informative and well written. If there is room, including some of the main points in the text should be considered.
- 4) Lines 206-211 discuss the differences between DE genes in female and male brains and leave us wondering what is to be made of these differences. A sentence or two to address this would be helpful.
- 5) On line 236, is there any indication or a way to determine what subset of neurons these are that express Ago3?
- 6) The novel class of Piwi positive cells is described as stem cells, based upon Dpn expression. Apart from their ability to proliferate, what other evidence is there that these are, in fact, stem cells? Do they further differentiate after proliferation? If so, they'll likely turn off Piwi and Dpn expression, correct? What is to be made of the fact that these stem cell markers remain on after proliferation in older flies?
- 7) Is there any indication from the differential expression data that neuronal plasticity is manipulated by the presence of Tud and glial granules?

Reviewer #3 (Remarks to the Author):

There seemed to be two stories in this paper, one about the role of Tud in dictating changes in the transcriptome based on its effect on transposons and noncoding RNAs and another story on the identification of a population of Dpn-positive cells in the adult brain that the authors posit are neural stem cells. I can see that, as this paper unfolded, the authors expected to find a population of stem-like cells in the adult brain given the expression of germ plasm proteins like Tud in the adult brain. However, I think that this expectation is misguided, and comes from a conflation of marker gene expression with stem cell identity that has led the authors towards biased conclusions that are not supported by their data. That said, there are important and potentially impactful findings presented in this paper that merit reframing, reinterpretation, and further investigation.

A strength of the paper is the work on the effects of Tud on transcription of transposons. Based on recent high profile and innovative efforts in neuroscience research, noncoding RNAs have emerged to be key regulators of neural development and neuronal physiology, and, in mammals, evidence has also emerged that transposons themselves also have important roles in aging-related neurological diseases. Using *Drosophila* to explore these two intertwined areas could yield truly innovative and impactful findings. Moreover, the role of transposons and noncoding RNAs in glial function in the aging brain is largely unexplored, and study of key regulators of these processes in glia should yield insights. There is a wealth of evidence that glia contribute to age related changes in the brain, and ample evidence that defects in RNA biology, particularly noncoding RNA biology, contribute to neurodegeneration. In this paper, the authors try to show that Tud, Ago3, Piwi, Vasa, and other proteins that regulate RNA biology in the germline are present in the adult brain, predominantly in cells that appear to be cortex glia. The authors also show that Tud mutants have transcriptional defects. However, the authors fail to show test the function role of these proteins beyond transcriptional profiling. What other phenotypes do these flies have? Did the authors try to knockdown Ago3, Piwi, and other germ plasm genes in glia and then monitor the animals? The authors mention that the expression patterns of these gene change with aging. Did they look for aging related phenotypes in Tud mutant animals, or in animals with glial specific knockdown of other germ plasm genes? Do such animals have neurodegeneration phenotypes? Behavioral phenotypes? Glial phenotypes? Synaptic plasticity phenotypes? Neuroinflammation phenotypes? Such experiments are the logical follow-up experiments to the data presented through Figure 3, and results from such experiments would potentially be impactful. The lack of such follow-up experiments and data is a missed opportunity.

In the last third of the paper, the authors claim that Piwi expression marks a set of adult stem cells in the fly brain. Figure 4 shows results that these cells co-express Dpn, which is used as a marker for *Drosophila* neural stem cells (neuroblasts). The authors also characterize the shape of these cells in Figure 5 and supplemental figures. However, at no point do the authors show that these cells are stem cells. A tissue resident stem cell is defined as such by its multipotency, which is comprised of an ability to give rise to a mixed lineage of differentiated daughter cells and maintain itself as a proliferative and undifferentiated state through self-renewal. At the core, a new stem cell population must be functionally defined, not defined by expression of marker genes associated with other stem cells. Marker genes can be expressed according to their biological function, not on their developmental role, and, as a result, marker gene expression alone is not sufficient evidence to define a new population of stem cells in the absence of functional data. No experimental evidence is presented that the Piwi-Dpn cells are stem cells. Such experimental evidence should come from adult lineage mapping or mitotic clones (Flp-FRT, etc), labeling for proliferation (BrdU, cell cycle gene expression, phospho-Histone H3 staining, etc), and evidence of self renewal and multipotent differentiation of daughter cells (lineages labelling coupled with staining for neuro-glial fate markers). Without such evidence the cells identified cannot be called a new population of stem cells.

Moreover, the shapes of the cell bodies shown and their projections shown are far more consistent with these Piwi/Dpn+ cells being neurons – they have unipolar asymmetrical cell bodies that are typical of adult *Drosophila* neurons and the projections are axons. The projections head towards midline structures that are known regions of neuropil, and in these regions there is fuzzy staining for Piwi that is likely from dendrites in the neuropil. In neurons, RNA binding proteins are well known to be trafficked in axons out to dendrites where they regulate localized translation in response to electrochemical cues, so I would expect Piwi to be trafficked in axons out to dendrites. Other investigators in the neuroblast and glial biology community have observed these large cell bodied Dpn-positive cells in the cortex of the adult brain, and, based on their anatomy and position, many investigators think that they are actually large cell bodied neurosecretory cells that send axonal projections towards the midline. The authors will need to do a series of co-stains to rule out whether the Piwi/Dpn+ cells are neurons in addition to the functional work to show that they are tissue-resident stem cells.

Other comments:

Also figures should be remade with color blind readers in mind in green and magenta.

Figure 1: The stains in (b) need to be shown singly in addition to being shown in overlay. In (c-h) could really benefit from dashed lines or additional dapi images to show the glial cell nuclei. This data would be much more convincing if the authors included a membrane marker for the glia, such as a membrane bound RFP.

Figure 3: The whole brain images could be shown in close-up, maybe with half brains shown, to allow readers an improved view of the cell morphology. The scale bars should be annotated with the relevant magnification in the figure proper. The Tud expression patterns are different between panels, and I can't tell if this is due to different magnification or section planes shown. There is no mention of parts H and I of the figure in the results text.

Figure 4-5: Co-stains with glial membrane markers, neuronal markers, and other neuroblast markers are needed to better support the authors' conclusions.

Figure 3: Why does the Tud expression patterns across these brain panels look so different. They should look the same. A re-occurring theme is that in all of the images, it is difficult to see the fluorescent signal. Is it possible to have a higher magnification insert panel and arrows to point specifically what you are referring to in the text.

Dear Dr. Lee

We appreciate your attention to our work and the reviewers' comments. We have now carefully considered the comments and revised the manuscript based on the reviewers' feedback. In particular, we toned down our original claim for the identification of bona fide stem cells throughout our manuscript and provided a more balanced interpretation of our data concerning unusual Piwi/Dpn-expressing cells in the adult brain. Also, we included new extended data fig1 panels d and e according to the suggestions of the referees regarding the data from experiments with original anti-Tud antibody and immunostaining control. Additional changes in our manuscript are highlighted in the text and our specific responses follow.

Reviewer #1:

'The work and findings for the role of Tudor in transposable elements repression and piRNA biogenesis are solid and very interesting. Specially the fact that the majority of brain piRNAs map to genes in the brain. They are sufficient for a short article and a good start for further investigations. In this case a discussion similar to the supplementary discussion would fit better.'

Supplementary discussion has been incorporated into the main discussion, also in accordance to Reviewer 2's comment.

'On the other hand, I find the evidence for calling the new and intriguing cell population identified "stem cell" weak. This statement is based on two facts: 1. the fact that these cells express a neuroblast marker and piwi but no extensive molecular characterization has been carried out, i.e other neuroblast or neuronal markers; 2. the number of cells increase with age. While this might be a property of certain stem cells, a strong evidence for calling these cells stem cells would be lineage tracing and seeing postmitotic differentiated cells derived from them. More work in this part could lead to an additional paper in the future.'

One way to maintain the two parts (Tudor results and new cell population) in the same paper would be to remove the term stem cell from the title, and rewrite intro and discussion accordingly. It is acceptable to hypothesize that they could be stem cells, as they do, but there is not enough evidence for using the term stem cell in the title. It would be misleading.'

"Stem cells" were removed from the title. Also, we toned down our claim of the identification of stem cells throughout our manuscript according to the reviewer's suggestion.

'Expression analysis of proteins in different glial types can be done with regular microscopy, is there any particular reason for using super-resolution in Figure 1?'

Tud was detected in small granules and to increase the resolution, we employed superresolution microscopy.

'What brain regions are the ones shown in Fig 1b to h? The images lack spatial context. It would be good to have a schematic showing which part of the brain is being imaged and its orientation.'

Fig. 1 was revised to improve presentation according to this and additional comments from the reviewer (see below). Area imaged in the Fig. 1 has been indicated in the schematic of the entire brain (now Fig. 1b) and the figure legend has been modified to accommodate for this change.

'The figure legend does not explain the two panels Fig1b. They look different, why? In the first Tud signal is specks and in the second Tud labels membranes? It is hard to relate this pattern to glia using a glial nuclear marker in this panel, the dots and membrane signal could be in neurons.'

The two panels in the original Fig1b were shown to indicate the distribution of Tud in relation to glial marker Repo. While in the right panel of this original figure Tud granules can also be seen, we agree with the reviewer that this is not as clear as it was in the left panel. Overall, the staining shown on the right panel was weaker and more diffused for both Tud and Repo channels than the image shown on the left, and, therefore, the image on the right gave the wrong impression that some Tud may be in the membrane. Consistent with the lack of Tud in the membrane are the superresolution images shown in Fig1g and h demonstrating lack of colocalization of Tud granules with the GFP-mCD8 marker that labels the membrane of perineurial and cortex glial cells respectively. Therefore, we removed the right panel in 1b in the revised figure and show image from the left panel as 1c.

We do not believe that some Tud is expressed in neuronal bodies embedded in the cortex glia. In fact, we never saw Tud in the brain outside of surface and cortex glia and Tud is also missing in the neuronal bodies in the cortex glia, which express Ago3 (Figs. 3f and g). In the revised manuscript, page 11, we put more emphasis on these data as follows: "Importantly, while Tud and Ago3 are found in the same granules in glia (Fig. 3f, Extended Data Fig. 6b, Supplementary Movie 1), Tud was never detected in the Ago3-positive neuronal bodies embedded in the cortex glia, consistent with glia-specific expression of Tud (Figs. 3f, g)."

'Material and methods indicate that detection of signal is done with antibodies. It would be informative to see a secondary control given that, since this a genomic reporter, lack of signal in the mutant background control cannot be performed.'

We included new images as Extended Data Fig. 1e, middle panels, which show brains' staining with anti-Fasciclin II antibody (α Fas II, marker for the mushroom body neurons) followed by the secondary Cy3-conjugated anti-

mouse antibody also used for detection of glia-specific expression of Flag-Tud. As shown in the Figure, there is no appreciable staining in Cy3 channel (red channel) anywhere, including Tud-expressing glia regions, except in the small area in the middle of the brain where the mushroom body neurons are selectively labeled by α Fas II.

'In the image provided for panel 1e it is difficult to see the spatial relationship between the red and green signal. Also it seems that the Tud-GFP (Fig 1b) gives a different pattern than Tud-Flag (Fig 1e).'

While the Flag-Tud staining shown in the original Fig. 1e (new Fig.1f) points to the punctate staining of Tud in cortex glia, similar to that of Tud-GFP in the original Fig. 1b (new Fig. 1c), we appreciate the reviewer's point comparing the two figures. However, original Fig. 1b is a superresolution image and original Fig. 1e is a low-magnification image obtained with 20X objective which cannot resolve individual Tud granules and the reason to show this low-resolution figure is to show the localization of Tud to a region of cortex glia. Accordingly, in the figure legend low-magnification images have been specified to distinguish them from the superresolution images.

In addition, we imaged with superresolution microscopy both Tud-GFP (new Fig. 1c) and Tud-Flag (Fig. 1g,h) and did not find difference in Tud distribution pattern and the size of Tud granules regardless which tag/antibody was used to detect Tud.

'And it is unclear what point wants to be made with panels Fig 1g and 1h. With the current image size it is hard to see well.'

Fig. 1g and 1h are the superresolution images which show that Tud is assembled into the granules in different types of glia. We included a statement on page 4 (Results' first section) that reads: "In addition, super-resolution microscopy showed the assembly of Tud into the glial granules (Fig. 1c, g and h)." Also, we made a reference to Fig. 1c,g and h when we characterize the glial granules in more detail in the Results' section titled "Germ-like glial granules". In addition, we increased the size of the Figs. 1g and h in the revised Figure 1.

'Are the Tud expression experiments in Fig 1 and Fig 3 done with homozygous Tud-tagged animals?'

Yes, this is correct.

'Have you noticed any behavioral defects in Tud mutant animals, given the presence of a significant number of DE expressed non-transposon genes? Maybe future experiments could include testing Tud mutants for different'

behavioral assays (<https://www.janelia.org/project-team/fly-olympiad>) in collaboration with Janelia Research Campus.'

We appreciate the reviewer's suggestion regarding the behavioral assays. We are planning this collaboration and hope these future behavioral experiments will provide further important insights into the function of *tud* and genes regulated by *tud* in the brain.

Reviewer #2:

'1) On line 58, *Tud* expression is examined in the brain with anti-*Tud* antibody. It would help if images of this staining and the westerns could be included in extended Figure 1.'

These data have been incorporated into the revised extended Fig. 1 (new extended data Fig1 d,e) as suggested.

'2) The paragraph beginning on line 121 that discusses the difference in piRNA biogenesis between the germline and brain could be expanded with references and something that addresses why this difference may or may not be expected. Then in the following paragraph, include an interpretation of the sense/antisense bias.'

The referred paragraph has been expanded (page 6). While we are not sure why there is a difference in the sense/antisense bias in brain and ovary piRNAs, we included an interpretation for the role of *Tud* in maintaining piRNA's sense bias in the brain in the Discussion (formerly supplementary discussion) on page 17th.

'3) Line 167 references the supplementary discussion. This supplement is very informative and well written. If there is room, including some of the main points in the text should be considered.'

Supplementary discussion is now a part of the main text (page 17).

'4) Lines 206-211 discuss the differences between DE genes in female and male brains and leave us wondering what is to be made of these differences. A sentence or two to address this would be helpful.'

Additional description and context/references have been provided as suggested (highlighted on page 10).

'5) On line 236, is there any indication or a way to determine what subset of neurons these are that express *Ago3*?'

We do not know what the identity of these Ago-3 expressing neurons is and this will be an interesting question to address in the future, both morphologically and functionally, which will require long-term efforts.

'6) The novel class of Piwi positive cells is described as stem cells, based upon Dpn expression. Apart from their ability to proliferate, what other evidence is there that these are, in fact, stem cells? Do they further differentiate after proliferation? If so, they'll likely turn off Piwi and Dpn expression, correct? What is to be made of the fact that these stem cell markers remain on after proliferation in older flies?'

We toned down our claim that these Dpn/Piwi-expressing cells are bona fide stem cells throughout the manuscript and removed "stem cells" from the title. In the revised manuscript, we indicate that although these unusual cells express both neuronal stem cell marker Dpn and Piwi, which is shown to be autonomously required for the maintenance of different stem cells both in germline and soma, more long-term detailed research will be needed to rule out that these cells are differentiated cell types.

'7) Is there any indication from the differential expression data that neuronal plasticity is manipulated by the presence of Tud and glial granules?'

There is no direct evidence from differential expression data that neuronal plasticity is affected by the presence of Tud. However, differential gene expression in *tud* mutants specifically related to the function of nervous system (Extended Data Fig. 4) is consistent with its potential effect on neuronal plasticity.

Reviewer #3:

'In this paper, the authors try to show that Tud, Ago3, Piwi, Vasa, and other proteins that regulate RNA biology in the germline are present in the adult brain, predominantly in cells that appear to be cortex glia. The authors also show that Tud mutants have transcriptional defects. However, the authors fail to show test the function role of these proteins beyond transcriptional profiling. What other phenotypes do these flies have? Did the authors try to knockdown Ago3, Piwi, and other germ plasm genes in glia and then monitor the animals? The authors mention that the expression patterns of these gene change with aging. Did they look for aging related phenotypes in Tud mutant animals, or in animals with glial specific knockdown of other germ plasm genes? Do such animals have neurodegeneration phenotypes? Behavioral phenotypes? Glial phenotypes? Synaptic plasticity phenotypes? Neuroinflammation phenotypes?'

As the reviewer indicated, there is a whole spectrum of the phenotypic tests that would be very interesting to do in addition to our datasets presented in the manuscript. As discussed above, a collaboration with Janelia Research Campus on many of these phenotypic tests would be a great start, and while we

are planning to do that, this will be a long-term undertaking and we believe, that this warrants another full study.

Regarding the aging phenotypes: the only aging-related phenotype reported in our manuscript is the increase in the number of the Dpn/Piwi+ cells during aging. Therefore, we do not have a reason/evidence to believe that testing of *tud* mutants or flies with glial specific knockdowns of other germline genes will yield aging-related phenotypes. However, these additional phenotypic tests, which specifically address the aging-related aspects, could also be attempted during the phenotypic testing with *Janelia*, which initially would be focused on just one age of the animals.

Reviewer's comments on additional work to demonstrate that Piwi/Dpn+ cells are stem cells and not differentiated cells (for example, large neurosecretory cells with axons).

We appreciate the reviewer's comments and toned down our claim throughout our manuscript that we identified a new population of stem cells. In the revised manuscript we point out that although we describe co-expression of Dpn (neuroblast marker) and Piwi in the same cells and Piwi is a well established stem cell factor autonomously required in germline stem cells and some somatic stem cells, we cannot rule out that these cells are in fact differentiated cells with unexpected and not previously found co-expression of Dpn and Piwi. While very extensive work will be needed to determine whether they are in fact stem cells or more differentiated cells (including lineage and clonal analysis), we feel that our current data can already be presented to scientific community and stimulate interest and promote further research on these unusual cells.

'Figure 1: The stains in (b) need to be shown singly in addition to being shown in overlay. In (c-h) could really benefit from dashed lines or additional dapi images to show the glial cell nuclei. This data would be much more convincing if the authors included a membrane marker for the glia, such as a membrane bound RFP.'

Original Fig 1b (now 1c) is a superresolution close-up image that corresponds to Fig.1a, which shows all single channels as well as overlay. The purpose of this Fig. 1b is to introduce Tud granules which can be clearly seen on this Figure without any obstruction from Repo which labels glial cells nuclei. Therefore, we feel that the additional single-channel Fig 1b images will be redundant and also, require some panels of the Figure to be smaller to stay within the current size. In addition, superresolution close-up image of DAPI/Tud staining in the cortex glia is shown in Fig. 3g. A membrane marker for the glia, GFP-mCD8, is presented in the superresolution images 1g and 1h that show no membrane localization of Tud.

'Figure 3: The whole brain images could be shown in close-up, maybe with half brains shown, to allow readers an improved view of the cell morphology. The

scale bars should be annotated with the relevant magnification in the figure proper. The Tud expression patterns are different between panels, and I can't tell if this is due to different magnification or section planes shown. There is no mention of parts H and I of the figure in the results text.'

And related comment on Figure 3:

'Figure 3: Why does the Tud expression patterns across these brain panels look so different. They should look the same. A re-occurring theme is that in all of the images, it is difficult to see the fluorescent signal. Is it possible to have a higher magnification insert panel and arrows to point specifically what you are referring to in the text.'

Figure 3 shows the distribution of subcellular glial granules co-labeled with the germ granule markers, and close-up Tud granule distribution in relation to different cell morphology markers (nuclei, membrane) is shown in Fig. 1 (for example, c, g, h) and also, in 3g (which shows a close-up superresolution segment of brain cortex stained with Tud, Ago3 and DAPI). Additional superresolution close-up images, which show clear fluorescent signals, are presented in Fig. 3e, f, as well as in superresolution sections of individual granules in 3h,i and a more extensive gallery of individual granules' superresolution images shown in the Extended Figures 5 and 6, as well as in Supplementary movies 1 and 2. In the text of revised manuscript, we now directly refer to these close-up images (including 3h and 3i) on page 10 to link the main text with these specific images/movies more effectively.

Scale bars on the figures are made in accordance with *Communications Biology* submission guidelines as follows. "Scale bars should be used rather than magnification factors, with the length of the bar defined in the legend."

Regarding Tud expression patterns in different panels in Fig. 3: we did not intentionally attempt to choose a similar section or similar overall intensity of the signal in the brains between these different experiments so while all experiments show a granular staining of Tud in the brain cortex and superresolution imaging detect similar sized Tud granules between different labeling experiments (for example, Fig 3e and f,g), Tud signal and Tud distribution shown in the panels are likely to come from different brain sections.

'Figure 4-5: Co-stains with glial membrane markers, neuronal markers, and other neuroblast markers are needed to better support the authors' conclusions.'

Figs. 4c and 4d show co-staining of Piwi and Dpn respectively with cortex glia membrane marker GFP-mCD8 and we modified the text that refers to this figures (page 11) and this figure legend to clearly specify that. We believe that additional stainings with other markers would be better suited as a part of a long-term follow-up study to comprehensively characterize these Piwi/Dpn+ cells by

multiple approaches described by the reviewer to determine whether these cells are stem cells or differentiated cells. As described above, we toned down our assertion on the identification of new stem cells and believe that work on the comprehensive characterization of these cells will need to be carried out during a different study.

On behalf of all authors, I would like to thank you and the reviewers for the work on our manuscript.

Sincerely,

Alexey Arkov

Reviewers' comments:

Reviewer #1 (Remarks to the Author):

The authors have satisfactorily addressed most of the concerns expressed in my initial revision.

However, I still think that there are more precise ways to demonstrate expression of tudor in cortex glia.

The authors choose two indirect ways:

1. absence of tudor signal in cortex glia membranes imaged by superresolution (Fig 1h) and assumption that expression is in glial cytoplasm
 2. absence of tudor signal in the cytoplasm of a subset of neurons that express Ago3 (Fig 3f,g) and assumption that expression in the area is in glial cytoplasm and no other neurons.
- Neither experiment demonstrates expression in cortex glia per se.

In principle, a very simple way would be to use a cortex glia driver and cytoplasmic GFP reporter or a combination of cytoplasmic and membrane tagged GFP reporters and analyze in that background the expression of Tudor-flag. Colocalization of flag signal with GFP would confirm glial expression, and maybe reveal as well expression outside that domain in some other neuronal types.

Minor comments

line 66 add (Extended Data Fig. 1a,b)

line 68 add (Extended Data Fig. 1c)

line 72 add (Fig. 1a)

line 263 might be a good place to indicate that while in neuroblasts *dpn* is expressed in the nucleus this new population of cells have expression in the cytoplasm.

Reviewer #2 (Remarks to the Author):

The authors have sufficiently addressed my previous comments. The paper and figures have been significantly improved. This is a substantial body of work that will have lasting impacts in its field.

Dear Dr. Lee

We have revised the manuscript to address a reviewer #1's comment on expression of Tud in cortex glia and details of our experiment are provided below.

Reviewer #1:

'The authors have satisfactorily addressed most of the concerns expressed in my initial revision. However, I still think that there are more precise ways to demonstrate expression of tudor in cortex glia.'

We have done an additional experiment which is described in a new extended data fig2 (see below). In this experiment, we used a new set of antibodies to allow for multi-labeling of cortex glia and imaging was carried out with Zeiss LSM 880 confocal microscope to generate conventional optical sections shown in this figure. In particular, we labeled cortex glia processes and membrane (with mouse anti-Wrapper antibody, green), cortex glia nuclei (guinea pig anti-Repo antibody, gray), Tudor granules (rabbit anti-FLAG antibody, red) and all nuclei (DAPI, blue; since cortex glia nuclei are labeled with Repo, all Repo-negative/DAPI-positive nuclei are those of neuronal bodies which are tightly outlined by cortex glia processes labeled with anti-Wrapper). We were able to confirm that Tudor granules are located in cortex glia, either within the cortex glia processes (Tud granules that are indicated with arrowheads) or close to cortex glia nucleus (Tud granules that are indicated with arrows).

Tud/Wrapper/DAPI Repo/Tud/Wrapper/DAPI

Extended Data Fig. 2 | Details of Tudor expression in cortex glia. Optical sections show cortex glial cells labeled with anti-Wrapper antibody (cortex glial cell membrane and cortex glial processes, green channel) and anti-Repo antibody (glial cell's nuclei, gray). DAPI stains nuclei and Repo-negative/DAPI-positive nuclei are those of neuronal bodies tightly surrounded by cortex glial processes labeled with anti-Wrapper. Tudor granules are labeled with anti-FLAG antibody (red channel). In different rows, the images are different optical sections from central brain cortex indicated in Fig. 1b and for a section presented in each row, right panel includes Repo signal in addition to that of Tud/Wrapper/DAPI shown in the left panel. Glial nuclei are indicated with “g” and neuronal bodies are indicated with dotted line and “n”. Tud granules are located either within the cortex glia processes (arrowheads) or close to cortex glia nucleus (arrows). Scale bars are 2 μm .

Also, we have revised our manuscript according to other comments of the reviewer #1 as follows.

'Minor comments

line 66 add (Extended Data Fig. 1a,b)

line 68 add (Extended Data Fig. 1c)

line 72 add (Fig. 1a)

*line 263 might be a good place to indicate that while in neuroblasts *dpn* is expressed in the nucleus this new population of cells have expression in the cytoplasm.'*

All done, corresponding changes are highlighted in the manuscript.

On behalf of all authors, I would like to thank you and the reviewer for the continuous attention to our manuscript.

Sincerely,

Alexey Arkov

REVIEWERS' COMMENTS:

Reviewer #1 (Remarks to the Author):

The additional experiment carried out by the authors clearly demonstrates the expression of Tudor granules in cortex glia.

All the comments in my previous review have been satisfactorily addressed.